# The Role of HDAC6 in Glioblastoma Multiforme: A New Avenue to Therapeutic Interventions?

**DOI:** 10.3390/biomedicines12112631

**Published:** 2024-11-17

**Authors:** Francesco Spallotta, Barbara Illi

**Affiliations:** 1Department of Biology and Biotechnology Charles Darwin, Sapienza University, 00185 Rome, Italy; francesco.spallotta@uniroma1.it; 2Istituto Pasteur Italia-Fondazione Cenci Bolognetti, Sapienza University, 00185 Rome, Italy; 3Institute of Molecular Biology and Pathology, National Research Council (IBPM-CNR), 00185 Rome, Italy

**Keywords:** glioblastoma multiforme, glioblastoma stem cells, epigenetics, HDAC6, epidrugs

## Abstract

Despite the great advances in basic research results, glioblastoma multiforme (GBM) still remains an incurable tumour. To date, a GBM diagnosis is a death sentence within 15–18 months, due to the high recurrence rate and resistance to conventional radio- and chemotherapy approaches. The effort the scientific community is lavishing on the never-ending battle against GBM is reflected by the huge number of clinical trials launched, about 2003 on 10 September 2024. However, we are still far from both an in-depth comprehension of the biological and molecular processes leading to GBM onset and progression and, importantly, a cure. GBM is provided with high intratumoral heterogeneity, immunosuppressive capacity, and infiltrative ability due to neoangiogenesis. These features impact both tumour aggressiveness and therapeutic vulnerability, which is further limited by the presence in the tumour core of niches of glioblastoma stem cells (GSCs) that are responsible for the relapse of this brain neoplasm. Epigenetic alterations may both drive and develop along GBM progression and also rely on changes in the expression of the genes encoding histone-modifying enzymes, including histone deacetylases (HDACs). Among them, HDAC6—a cytoplasmic HDAC—has recently gained attention because of its role in modulating several biological aspects of GBM, including DNA repair ability, massive growth, radio- and chemoresistance, and de-differentiation through primary cilia disruption. In this review article, the available information related to HDAC6 function in GBM will be presented, with the aim of proposing its inhibition as a valuable therapeutic route for this deadly brain tumour.

## 1. Introduction

The high number of clinical trials (2003 on 10 September 2024) attempting, at least, to increase the life expectancy of patients affected by glioblastoma multiforme (GBM) gives an idea about the effort the scientific community is providing to identify valuable therapies for this neoplasm, which is still neither defeated nor curable. Indeed, whereas living with an increasing number of other cancer types is now possible, a GBM diagnosis is a death sentence in about 15–18 months [1].

The identification of whatever molecule or strategy able to slow or even eradicate any kind of tumour requires a deep understanding of the biology and molecular and metabolic processes that make that specific tumour a unique entity with specific vulnerabilities. A great body of knowledge has been acquired for GBM in the past few years. OMIC studies, even at the single-cell level, shed light on the molecular pathways working in glioblastoma cells, helping with the classification of GBM into subtypes [2,3,4] and leading to testing a variety of new potential therapeutic strategies, including (i) the administration of monoclonal antibodies, small molecules, and protein kinase inhibitors and (ii) the application of electric fields, intraoperative photodynamic therapy, and CAR-T immunotherapy (see the clinicaltrials.gov website). However, none of these routes have been proven to be really effective in treating this brain cancer, which is still faced with maximal surgical resection, radiotherapy, and temozolomide (TMZ)-based chemotherapy.

GBM may be classified into the proneural, classical, and mesenchymal subtypes [2]. This classification takes into account mostly the transcriptional profile typical of each subtype, which is mirrored by the transcripts output of the glioblastoma stem cells (GSCs) from which each tumour stems. The discovery of GSCs [5] was a milestone, both from a scientific and a clinical point, because, from then on, they represent the major therapeutic target for GBM. Indeed, GSCs are responsible for GBM recurrence and resistance to therapies, as their surgical eradication is always incomplete and engage a series of molecular pathways, including DNA repair mechanisms, to overcome the cell damage occurring upon exposure to radiations and chemotherapy [6,7,8,9]. By definition, GSCs hold the characteristics of self-renewing pluripotent stem cells. A small percentage is also capable of asymmetrical cell division, which gives rise to cycling stem cells and differentiated cells which increase the tumour mass [10,11,12]. As stated above, GSCs mirror the biological, molecular, and genetic characteristics of the tumour subtype they belong to, indicating the requirement of differential therapeutic approaches according to the subtype-specific vulnerability or the identification of a common player, pivotal to GSCs survival, to be explored as a potential candidate for GBM treatment.

Typical epigenetic abnormalities are frequently found in GBM and, as in other tumours, the main outcome is both the silencing of tumour suppressor genes and the activation of DNA damage repair systems, among others [13]. These phenomena depend on the modification of the DNA methylation profile, the aberrant DNA methylation–demethylation cycle, and changes in both the histone modification pattern and chromatin architecture [14] within the regulatory regions of the above-mentioned genes. Each of these processes is ruled by specific enzymes that are variably either overexpressed or downregulated in GBM (see Section 3 below). Among them, histone deacetylases (HDACs) have been found to be heavily deregulated in GBM. Members of class I HDACs (HDAC 1, 2, 3, and 8) are mostly overexpressed. Class IIa HDACs (specifically HDAC 4, 5, and 9) are frequently upregulated, while, among class IIb, HDAC10 is downregulated and HDAC6 is upregulated. Sirtuins, belonging to class III, may be both up- or downregulated [15]. The unique member of class IV HDACs, HDAC11, has been found to decrease along GBM’s progression [16]. While a considerable number of preclinical studies report the use of HDAC inhibitors [15] and underly the importance of HDACs’ deregulation in GBM malignancy, few clinical trials (22 on 10 September 2024) employed or are employing HDACi, usually as a combo-therapy with TMZ and radiations, suggesting that the use of these small molecules, also called epidrugs, still has to be implemented, and further studies are needed to identify the best therapeutic candidate.

In 2020, the Society of Neuroncology (SNO) and the European Society of Neuroncology (ESNO) provided an extensive consensus review on the current status of knowledge and management of primary GBM [17]. On the front of therapies, the most recent strategies and hypotheses of targeted interventions are reported, including the use of Bevacizumab (not approved by the European Medicines Agency for clinical use), nistrosoureas (lomustine, carmusitne, and fotemustine, the latter of which is not approved by the Food and Drug Administration), and kinase inhibitors already used for other cancer types. Except for a certain degree of efficacy of lomustine in prolonging the overall survival (OS; 7.1–8.6 months) and progression-free survival (PFS; 1.5–3 months), none of these treatments were proved to be efficient. The high heterogeneity of GBM is probably the basis of these failures. Immunotherapy is at the forefront of GBM treatment. Stimulating responses to tumour antigens and overcoming microenvironment immunosuppressive mechanisms and immunotolerance are pivotal to GBM eradication. Different immunotherapeutic approaches are currently ongoing, including chimeric antigen receptor T cells (CAR-T) transfer, immune checkpoint inhibitors (ICs), oncolytic viruses (OVs), and cancer vaccines (CVs). All of these routes showed to be promising, but further refinements and trials are needed [18].

HDAC6 is currently a hot topic in the GBM field. The most recent literature underscores an increasing number of signalling pathways and molecular processes that are affected and controlled by this protein in GBM cells. Significantly, HDAC6 is overexpressed in GSCs, contributing to the maintenance of their stemness and the resistance to chemotherapeutic agents [19,20]. Specifically, HDAC6 has been found to promote proliferation and resistance to TMZ in GBM cells [21,22,23]. This latter phenomenon depends on its ability to modulate DNA damage repair [20,24,25]. Further, it restricts GSCs differentiation through Sonic Hedgehog (Shh) pathway activation [26,27] and primary cilia disassembly [19], while inducing the epithelial-to-mesenchymal transition (EMT) and autophagy [28].

In light of this evidence, HDAC6 inhibitors (HDAC6i) are now at the forefront of GBM treatment, with some of them having already entered into clinical trials.

In this work, a detailed overview of the aforementioned HDAC6 functions in promoting and sustaining GBM malignancy will be provided. Further, the possibility of employing its pharmacological inhibition as a novel, adjuvant strategy to fight GBM will be discussed.

## 2. Glioblastoma Multiforme

Primary (i.e, de novo) GBM is a rare brain tumour and is defined as a grade-IV astrocytoma, distinct from secondary GBM arising from low-grade (I-III) astrocytoma. Despite its rarity, GBM represent 14.3% of all the tumours affecting the central nervous system (CNS) and 49% of malignant CNS cancers in adults (>50 years of age [29]). However, GBM may also occur in children with distinct genetic characteristics [30].

Before GBM classification into molecular subtypes, histopathological variants have been defined by the World Health Organization (WHO), including gliosarcoma, GBM with giant cells, and conventional GBM. Additional histotypes were identified later [31], underlying the high heterogeneity of this cancer and explaining the term “multiforme”, but the implementation of transcriptomic analyses definitely locked within specific molecular boundaries the GBM subtypes that define the currently accepted classification [2].

### 2.1. GBM Classification: From (Epi)genetics to Transcriptomic Profiles

GBM classification has been thoroughly discussed in [32]. Here, a concise summary of the concepts and steps that led to the actual GBM classification will be provided.

In addition to the histological characteristics, genetic abnormalities represent the first level of GBM classification. Primary and secondary GBM may be discriminated, for example, by the frequency of genetic alterations, rather than the type [33], and among them, the point mutations in the enzyme encoding isocitrate dehydrogenase (IDH) 1 and 2 were identified as robust markers for these tumours [34]. Indeed, 70% of secondary GBMs carry the R132H mutation in IDH1, which is present only in 5% of primary GBM [35]. The R172K mutation in IDH2 is very rare and is present in only 3% of secondary GBMs, oligodendrogliomas, and grade II-III astrocytomas [35]. Other frequent genetic abnormalities found are the loss of heterozygosity (LOH) of chromosome 10q or the monosomy of chromosome 10, epidermal growth factor receptor (EGFR) gene copy number gain on chromosome 7, deletion of cell cycle inhibitor genes, such as p16INK4a and cyclin-dependent kinase inhibitor (CDKN) 2A and 2B, and TP53 and PTEN mutations [36]. EGFR amplification is typical of highly aggressive primary GBM; about 50% of these tumours present a truncated and constitutively active form of EGFR (EGFRvIII), lacking part of the extracellular domain as a result of exons 2–7 deletion [37]. Other typical mutations are found in genes encoding for CDK4 and 6, hepatocyte growth factor receptor (MET), and murine minute doubles (MDMs) 2 and 4 [35,38]. Secondary GBM may also carryATRX mutations [30].

Chromatin organisation may provide a further level of classification, as changes in epigenetic modifications may parallel genetic abnormalities, although this is not a rule. While pediatric GBM presents mutations in histone-coding genes [39,40], adult GBM mostly present a typical DNA methylation pattern [41] at the promoter regions of both tumour suppressor genes (hypermethylation) [42,43,44] and genes deputed to cell proliferation, escape from apoptosis, and DNA repair (hypomethylation) [45]. In addition, this modification relies on DNA methyltransferase activity [46]. Abnormalities in the methylation profile of GBM cells also depend on the deregulated activity of DNA demethylating enzymes, namely the ten-to-eleven translocation (TET) family of proteins [47], which typically occurs in GBM carrying IDH1 mutations [48,49,50]. Indeed, mutated IDH1 transforms isocitrate into D-2-hydroxy-ketoglutarate (D2-HG), which inhibits the dioxygenase activity of TETs, leading to an accumulation of methylated cytosine residues and the occurrence of the so-called glioma hypermethylator phenotype (G-CIMP) [51,52]. In addition, IDH-mutated GBMs present disorganised topological associated domains (TADs), three-dimensional chromatin structures that regulate gene transcription controlling the proximity of regulatory regions, mainly enhancers, with gene bodies [52]. In low-grade gliomas, the N6-methyladenosine (m6A) profile of specific miRNAs has been found to differentiate between low- and high-risk gliomas more than IDH mutation [53]. This aspect will be not further discussed here. For more information, see [32].

In the early 2000s, the first large-scale studies opened the way for the identification of the transcriptional signatures that are able to discriminate between astrocytomas and GBM and, within GBM, among three different subtypes, whose molecular and biological characteristics have been refined over the years.

Microarray studies led first to the identification of 360 genes able to differentiate astrocytomas and GBM [54] and then to the discovery of 35 transcripts specific for three different GBM subtypes [55]. Most of these genes are related to proliferation, migration, and, more importantly, to key neurogenetic steps, supporting the hypotheses that GBM may arise from the re-activation of normal developmental routes in quiescent stem cells resident in the brain [56] or from de-differentiation processes from astrocytes to neural stem cells that derail into GSCs [57,58]. Thus, the first classification of molecular GBM subtypes defined the proliferative (Prolif), mesenchymal (Mes), and proneural (PN) subtypes mirroring the biological and molecular properties of these tumours, with the Prolif subtype characterised by the expression of genes controlling the cell cycle, the aggressive Mes subtype marked by genes involved in neo-angiogenesis, and the PN subtype associated with a more differentiated phenotype, due to the expression of neural differentiation-related genes [59,60,61,62]. Each subtype-specific panel was further associated with genetic lesions [38], detailing further GBM subtypes at the molecular level till the definition of the current nomenclature used to classify GBM into classical, mesenchymal, and proneural subtypes [2]. A fourth subtype, called neural and thought to be the result of a contamination of normal brain cells within the samples undergoing RNA sequencing [4], has been recently validated and is the result of the activation of neural developmental programs in GSCs located at the tumour border by signals from surrounding neurons [63].

Classical and mesenchymal subtypes consistently overlap at the transcriptomic level, but the mesenchymal subtype also presents an inactivation of neurofibromin 1 (NF1), overexpression of the YKL-40 gene, and proteins of NF-κB network, which is consistent with the inflammatory features that characterise this tumour. Furthermore, this subtype is associated with the transcripts belonging to tumour-associated glia and microglia, indicating a strict communication between the tumour and the surrounding microenvironment and the possibility of tumour re-shaping according to tissue environmental signals [4]. The proneural subtype invariably bears platelet-derived growth factor receptor alpha (PDGFRA) alterations and hot-spot mutations in IDH1. Moreover, genes involved in oligodendrocyte differentiation are also expressed [2].

A hypermutator phenotype was also uncovered. These GBMs belonged to a cluster of cancers treated with TMZ or lomustine or both and presented mutations in the DNA mismatch repair system and heavy methylation of O-6-methylguanine-DNA methyltransferase (MGMT) promoter [64]. MGMT removes the alkyl residues added by chemotherapeutic agents to the DNA. Therefore, MGMT promoter methylation represents both a prognostic marker and, when hypomethylated, indicates tumours resistant to chemotherapeutic approaches [65,66].

Notably, a shift from the proneural to the mesenchymal subtype has been observed upon tumour recurrence and is associated with the expression of the YKL-40 gene, encoding for a glycoprotein of the extracellular matrix [59,60,61,62].

### 2.2. Glioblastoma Stem Cells (GSCs)

As with most tumours, GBMs possess tumour-initiating cells, named glioblastoma stem cells (GSCs) [67]. Although representing a minority of tumour cells, CD133^+^/CD44^+^ GSCs are the real cells that are responsible for GBM aggressiveness and infiltration, due to the activation of neoangiogenesis, resistance to therapies, and finally, the death of patients, as their complete surgical eradication is not possible without causing severe brain damage. Thus, over the years, they became both a challenge and a major target for scientists and neurooncologists.

GSCs closely resemble normal neural stem cells that are capable of self-renewal due to the activation of the Nanog-Sox2-Oct4-KLF-4 transcriptional circuitry and unlimited proliferative ability due to Myc oncogene overexpression [68,69,70,71]. The Sonic Hedgehog (Shh), notch, and Wnt/β-catenin signalling pathways are also active in GSCs but are often heavily deregulated, leading to a transcriptional derailment of the genes involved in the proliferation and differentiation of neural stem cells [7,72,73,74]. As we will see below, abnormalities in these pathways also depend on the lack of primary cilia, which characterises terminally differentiated neural cells.

It is widely accepted that GSCs recapitulate at the phenotypic and genomic level the tumour subtype of origin. However, in surgical GBM explants, mostly Mes and PN GSCs are found [75,76]. They are believed also to be responsible for GBM intratumoral heterogeneity [77,78], but this phenomenon does not depend on (epi)genetic inheritance in the tumour progenies but rather on growth and cell fate determinants [79]. Among them, the interaction with cells, especially immune cells, in the tumour microenvironment represents a fundamental source of GBM heterogeneity and evolution. Indeed, each GBM subtype-specific transcriptome is typically associated with transcriptional signatures of different immune cell populations. Furthermore, these immune signatures may evolve and change upon the acquisition of a resistant phenotype. Moreover, the genes involved in normal neural development have been detected in GSCs at the outer border of the tumour core as a result of signals from neighbour neurons in brain regions not invaded by cancer cells [4].

#### 2.2.1. GSCs and Resistance to Radio- and Chemotherapy

Resistance to GBM gold-standard therapies, namely radiations and TMZ, is a unique property of GSCs, as tumour cells lacking the CD133 stem marker were proven to be not resistant [12]. To achieve both radio and TMZ resistance, GSCs activate diverse DNA repair systems. Indeed, although both CD133^−^ and CD133^+^ GBM cells activate the ataxia-telangiectasia-mutated (ATM), Rad17, Chk1, and Chk2 checkpoint proteins, only CD133+ GSCs potently phosphorylates Chk1 and Chk2 [8]. Other efficient DNA repair mechanisms deployed by GSCs include the single-strand break repair (SSBR) system, activation of poly (ADP-ribose) polymerase (PARP) [6,80], and activation of MGMT due to its promoter hypomethylation [55,56]. Intriguingly, the NF-κB-dependent PN-Mes shift [45] also favours GBM resistance [9], as the cells belonging to the Mes phenotype have been demonstrated to show less γH2AX foci, suggesting efficient DNA repair, and do not accumulate in the G2/M phase of the cell cycle [45].

Specific epigenetic modifications have been shown to be associated with the TMZ resistance of GSCs. In particular, downregulation of the H3.3 histone variant has been demonstrated to discriminate between resistant and nonresistant cells [79,81]. Indeed, histone H3.3 favours the differentiation of GSCs that exploit the mixed-lineage leukaemia 5 (MLL5) methyltransferase to silence its expression [81].

#### 2.2.2. GSCs and the Primary Cilium

The primary cilium is a non-motile, microtubule-based organelle protruding from the cell membrane, serving as an “antenna” for the extracellular signals ruling developmental processes and cellular homeostasis [82]. Its central core is constituted of more than 1000 polypeptides (ref. [82] and https://esbl.nhlbi.nih.gov/Databases/CiliumProteome/, accessed on 15 October 2024), making the primary cilium one of the most complex cell organelles. Typically, the primary cilia are present in terminally differentiated cells and are assembled during the G0/G1 phase of the cell cycle [82]. The importance of the primary cilium in the maintenance of proper cellular development and homeostasis is highlighted by a group of genetic diseases called “ciliopathies”, characterised by a wide range of mutations in many different ciliary genes, with a profound impact on the CNS (especially the cerebellum), kidney, eyes [83,84]. It is not surprising that primary cilia loss occurs in several types of cancer (breast, pancreas, and kidney). Representing a brake for cell cycle progression, tumour cells have to devise strategies leading to cilium disassembly.

##### Brief Overview of the Structure of Motile and Non-Motile (Primary) Cilia

Historically, the structure of motile cilia was found to be constituted by nine microtubule doublets that surround a couple of central microtubules (9+2 axoneme), whereas primary, non-motile cilia have a 9+0 structure [82]. However, a 9+4 axoneme has been identified in rabbits and a 3+0 structure in protozoans [85,86]. Of note, microtubules within the cilia axoneme have to be acetylated to allow the proper assembly and function of the cilium. These hairy-like organelles originate from a basal body formed by two centrioles. The mother centriole represents the cilium root at the cell membrane. The whole basal body is not only the centre of cilium assembly/disassembly but also controls protein entry and exit within the cilium, being in strict connection with the transition zone, a region which houses many proteins that are later transported to the ciliary membrane [87,88,89]. This latter holds many receptors, transporters, ion channels, and sensory proteins, while the soluble cilioplasm contains signalling molecules. Intraflagellar transport (IFT) motor proteins, which “walk” on the axoneme microtubules, allow the anterograde and retrograde transport of molecules within the cilia [90], whereas Bardet–Biedl syndrome (BBS) proteins (also called the BBSome) function as adaptors between the IFT machinery and the ciliary membrane, allowing for the compartmentalisation of molecules within the cilia [91].

##### Altered Primary Cilia Signalling in GBM

Primary cilia are the collectors of a wide variety of signalling pathways to a greater extent than thought before [92]. Within the scope of this work, we will discuss the altered cilia-dependent signalling pathways that are relevant to GBM and GSCs biology.

One of the most important primary cilium-dependent pathways is Shh signalling, which controls the activation of many transcriptional programs, including the differentiation of neuroblasts, cell polarity and patterning, and when deregulated, tumorigenesis. The final players of the Shh cascade are the glioma-associated oncogene homologue (Gli) transcription factors (TFs) [93,94,95] located within the cilium. In the absence of Shh, the suppressor of fused (Sufu) protein binds and inhibits the nuclear translocation of the Gli family. On the contrary, when Shh is produced, it binds to its receptor PTCH1, which is removed from the cell membrane, releasing the smoothened (SMO) receptor to the cilium where it inhibits the conversion of active Gli transcription factors into repressor forms [92]. In GBM, where the Shh pathway is overactive, Gli proteins promote the transcription of genes involved in stem cell maintenance, tumour progression, and resistance to therapy [73,96];Centrosomes play a pivotal role in orchestrating the nucleation of interphase microtubules, are involved in different signalling pathways, and control cell motility and ciliogenesis. The core of the controsome is represented by the centriole, which organises the surrounding pericentriolar material provided with centriolar satellites, whose major component is the pericentriolar material 1 (PCM1) protein [97]. PCM1 has many functions, including the correct location of centrosomal proteins in the centrosomes. Therefore, PCM1 dysfunction has a remarkable impact on cell physiology and is associated with both primary cilium-related diseases and cancer. Although a functional PCM1 is related to the proper assembly of the cilium (i.e., to a “non-cancer state”), in GBM, it has been reported that PCM1 depletion—and, consequently, primary cilia deprivation—is favourable to GBM cell death and TMZ sensitivity [98];The cell cycle-related kinases (CCRK)–intestinal kinase (ICK)–male germ cell-associated kinase (MAK) pathway negatively regulates ciliogenesis, while playing a pivotal role in intraflagellar transport [99,100,101] and Shh response [102]. CCRK depletion in both fibroblasts and GBM, which presents high levels of CCRK, results in cilia restoration, as a lack of phosphorylation of the CCRK downstream effector ICK inhibits its suppressive effect on ciliogenesis. Consequently, CCRK and ICK depletion produce cell cycle arrest in a cilia-dependent manner [103].

##### Ciliogenesis in GSCs

The presence and role of the primary cilia in GSCs are controversial. Goranci-Buzhala et al. [104] reported that most of the GSCs do not possess primary cilia, irrespective of their specific subtype of origin, and show accelerated proliferation. This characteristic, typical also of neural precursor cells (NPCs), depends on the high expression of the proteins forming the cilium disassembly complex (CDC), which includes HDAC6 (Figure 1). Depletion or inactivation of one CDC component—the Nek2 kinase, important for the activation of kinesin 24 (Kif24), which depolymerises microtubules—results in cilia restoration and GSC differentiation, but only in the proneural subtype, indicating that putative CDC targets are subtype-specific. Indeed, cilia-dependent differentiation of PN GSCs depends on PDGFRα relocation to the primary cilia [104] Indeed, PDGFRα, that is altered in the PN subtype, is widely diffused in unciliated GSCs [2]. Thus, in GBM, the presence and the function of primary cilia are probably subtype- and context-specific. This phenomenon further confirms the heterogeneity of this tumour and its resident stem cells. More recently, it has been shown that 25% of GSCs possess primary cilia. In these cells, Sox2-dependent Kelch domain-containing 8A (KLHDC8A) is required for primary cilia formation, Shh signalling, and GSC proliferation, whereas KLHDC8A deprivation leads to cilia loss and decreased in vivo tumorigenic properties of GSCs [105]. Further, the multidomain protein BAG3, involved in the suppression of ciliogenesis and cilia homeostasis, is overexpressed in GBM and participates in GSC stemness maintenance [106]. Similarly, ciliogenesis impairment, obtained through PCM1 depletion, results in GSCs’ decreased proliferation, induction of apoptosis, and sensitivity to TMZ [98].

## 3. Epigenetics of GBM

Rearrangements of chromatin structure and topology in cancer cells, including GBM cells, frequently lead to the aberrant regulation of transcripts normally ruling basic cell properties, such as proliferation and death, migration, and differentiation.

### 3.1. DNA Methylation

The most frequent epigenetic abnormality found in GBM is represented by changes in the DNA methylation pattern, mostly at the CpG islands of gene promoters. Methylation of cytosine at position 5 (5mC) is highly stable and, thus, heritable. De novo methylation is catalysed by DNA methylatransferases 3A and 3B [107,108], whereas methylation pattern maintenance across cell generations is ensured by DNA methylatransferase 1 [109]. Although typical of CpG islands, 5mC is found also in non-CpG genomic regions, especially in highly differentiated adult cells, including neurons [110,111,112]. At the epigenetic level, 5mC content at gene promoters has been definitely associated with gene repression. However, 5mC is highly enriched in the bodies of actively transcribed genes [113], specifically in exons compared to introns. Two possible mechanisms have been proposed to explain this paradoxical role of DNA methylation, namely the regulation of mRNA splicing [114,115,116,117] and the repression of intragenic, cryptic promoters [118]. Furthermore, an activating role for the methylation of CpGs within promoters has been also documented [119].

In GBM, the most well-known and characterised gene affected by DNA methylation at its promoter is MGMT, which has a highly prognostic value [120]. By removing alkyl residues from TMZ-modified DNA, its expression strongly predicts patient outcome. Thus, hypomethylation of MGMT promoter results in worst prognosis and reduced median survival, as TMZ is ineffective. Usually, 5mC residues within chromatin act as docking sites for methyl CpG-binding domain (MBD) proteins, which, in turn, recruit other classes of transcriptional co-repressors, such as histone methyltransferases [121]. It has been reported that the polycomb repressor complex (PRC) member enhancer of zeste homologue 2 (EZH2) is capable of conveying DNA methyltransferases onto target regulatory loci, strictly linking DNA methylation with repressive histone modifications and creating highly compacted chromatin regions [122]. Aberrant DNA methylation profiles may also occur upon the dysfunction of demethylating enzymes, such as the TET family of proteins. TETs transform 5mC into unmethylated cytosine through a series of oxidative steps that foresee the formation of 5-hydroxymethylcytosine first (5hmC), 5-formilcytosine (5fC), and 5-carboxylcytosine (5caC) [123,124]. The latter is excised by thymine DNA glycosylase (TDG), leaving an abasic site that is later replaced by the base excision repair (BER) system [125,126]. The IDH1 mutant GBM typically presents a deregulated methylation–demethylation cycle, due to the production of 2D-HG by mutant IDH1 from isocitrate. The 2D-HG cannot serve as a co-factor for TETs, which are inhibited, with the consequent accumulation of 5mC and 5hmC and the occurrence of the so-called glioma CpG methylator phenotype (G-CIMP). This phenotype has been recently found to be also correlated with 5hmC content. The loss of 5hmC corresponds to 5mC enrichment at the chromatin loci, which also present high levels of dimethylated histone H3 on lysine 4 (H3k4me2), providing a novel glioma-specific chromatin signature [127,128]. Specific DNA methylation patterns may alter the binding of TFs and the related co-regulators (both activators and repressors) and even the chromatin architecture (for details see [129]).

### 3.2. Histone Acetylation

A wide variety of histone modifications have been identified to date, including acetylation, methylation, isomerisation, sumoylation, ubiquitilation, nitration, carbonylation [130], lactylation [131], and glutationylation [130]. According to the scope of this review manuscript, we will focus on acetylation/deacetylation of histones.

With the exception of histone H4 acetylation on lysine 20 (H4K20ac), associated with gene repression [132], the addition of acetyl moieties to the K residues at the N-terminal tail of histones is, probably, the most well-established epigenetic modification leading to transcriptional activation. The presence of high levels of histone acetylation in GBM specimens compared to healthy controls, and the further enhancement along the tumour stages, remarks upon the importance of this modification to the onset and progression of this neoplasm [133]. Specific histone acetylation profiles have been recently related to the single-cell types present within GBM. Thus, cells with high levels of histone acetylation have been found to belong to the oligodendrocyte precursors and myeloid lineages [134], which is consistent with the activation of immune cells in the tumour microenvironment [133,134]. Enrichment analyses showed that histone acetylation is linked to the activation of genes involved in focal and cell adhesion, actin binding, axon development, and differentiation programs [134]. The relative level of histone acetylation has been also used to design a risk model that proved to be as effective as those currently used. High- and low-risk subgroups have been identified, which are able to predict the clinical outcome of patients with overlapping accuracy compared to the other risk models. Acetylation levels have been also related to the efficacy of immuno- and chemotherapy [133]. Moreover, a recent work revealed that, in GSCs, high levels of simultaneous H4K8ac and H4K5ac identify superenhancers that are involved in the regulation of oncogenes pivotal for stemness maintenance [135].

### 3.3. HATs and HDACs in GBM

Acetylation, as well as all the other epigenetic modifications, is reversible. Histone acetyltransferases (HATs) and histone deacetylases (HDACs) are the enzymes that catalyse the addition and the removal, respectively, of histone K acetyl groups. Both are deregulated, to different extents, in GBM.

#### 3.3.1. HATs Deregulation in GBM

HATs are divided into three subgroups, namely (i) the Gcn5-related N-acetyltransferase (GNAT); (ii) the MOZ, Ybf2-Sas3, Sas2, the Tip60 (MYST); and (iii) the CREB binding protein (CBP)/p300 family. Acetylating also a variety of non-histone proteins, such as TP53, Myc, NF-κB, HATs may impact gene transcription either directly, through histone modification, or indirectly, inhibiting or potentiating the activity of the protein they acetylate. Thus, it is not surprising that the deregulation of HATs has been observed in many human diseases, including cancer. Remarkably, the role of HATs in GBM is poorly investigated. It is known that p300 has a tumour suppressor role in GBM, inducing the differentiation of GBM cells into astrocytes and inhibiting migration/invasion [136]. Indeed, the EID3 p300 inhibitor is overexpressed in GBM [137]. Another paper indicated KAT6A as a putative GBM molecular target, as its overexpression activates the PI3K/Akt pathway, leading to increased tumorigenic properties [138]. In rodent models of glioma, KAT8 activity, which mediates H4K16 acetylation, has been related to microglia conversion from an immune tumour counterpart to a tumour-supporting layer. KAT8 recruitment at the microglial gene promoters is Sirt1-dependent, providing an interesting example of cooperation between HATs and HDACs in promoting tumorigenesis [139].

#### 3.3.2. HDACs Deregulation in GBM

Contrary to HATs, HDACs are known to be heavily deregulated in GBM.

Class I HDACs are ubiquitously expressed, although they may have peculiar roles within cells. Indeed, whereas HDAC1 and 2 are involved in the control of cell proliferation and neurogenesis [140], HDAC3 is associated with lipid metabolism [141] and cardiac [142] and neural development [143], while HDAC8 controls sister chromatid exchange, muscle contractility, microtubule stability, and energy homeostasis [144]. All of them are overexpressed in a wide variety of cancers [15]. In GBM, HDAC 1 and 2 depletion have been demonstrated to reduce GBM cells’ proliferation and migration while inducing TMZ sensitisation [145,146]. HDAC3 is mainly overrepresented in aggressive glioma phenotypes and is an unfavourable prognostic marker [147]. HDAC8 is also associated with a block of the cell cycle and decreased levels of MGMT, thus promoting TMZ sensitisation [148];Class II HDACs are divided into two subclasses, a and b. Subclass IIa HDACs (4, 5, 7, and 9) are highly tissue specific. HDAC5 and 9 are expressed in neurons and the heart [149]. HDAC4 is mostly expressed in bone [150], neurons [151], and in cells of mesodermal origin, as well as HDAC7 [152]. These HDACs are provided with nuclear localisation (NLS) and nuclear export (NES) signals that control their intracellular location, according to a variety of stimuli [153]. Subclass IIb (HDAC6 and HDAC10) has a mostly cytoplasmic localisation and selective substrate specificity [154,155]. Indeed, HDAC6 deacetylates mainly non-histone proteins, such as tubulin, cortactin, and geat shock protein (HSP)90 ([156], see Section 4), while HDAC10, a polyamine deacetylase, shows selectivity for long *N*8-acetylspermidine, acetylputrescine, and acetylcadaverine [155]. As well as class I HDACs, both class IIa and class IIb HDACs are deregulated in neoplastic malignancies and GBM. Class IIa HDACs are overexpressed in GBM cells and are associated with enhanced proliferation and migration/invasion [15], as well as class IIb HDAC6 [28]. HDAC10 is downregulated in paediatric GBM and astrocytomas [16], but no information are available to date regarding adult GBM;The peculiar seven members of class III HDACs (also called Sirtuins) are NAD^+^-dependent and do not possess a Zn^2+^ binding domain. They are located in many cellular compartments, including the cytosol, nucleus, and mitochondria [157,158,159,160]. Although histones are deacetylated by Sirtuins, these enzymes exert their activity mainly on non-histone substrates, regulating an impressive number of biological processes with a profound impact, especially on metabolism, oxidative stress, and energy homeostasis. Therefore, the deregulation of Sirtuins is found in many human disorders, from cardiovascular diseases to neurodegeneration, from inflammatory disorders to cancer. SIRT1,2,3,6 and 7 have been found to be deregulated in GBM, with conflicting results. SIRT1 has been identified both as an oncogene [161] and a tumour suppressor in GBM cells [162]. SIRT2 is a tumour suppressor in GBM cells and acts through the NF-κB/p21 pathways [163]. SIRT3 has a prognostic value. The higher the SIRT3 expression levels, the worse the life expectancy [164]. SIRT6 inhibits both proliferation and metabolic reprogramming in GBM cells [165], whereas SIRT7 enhances the proliferation of glioma cells via the ERK/STAT3 axis [166];HDAC11, representing the sole member of class IV HDACs, is tissue-specific and expressed in a variety of organs, including the heart, kidney, and brain. Its localisation varies from the cytoplasm to the mitochondria to the nucleus, according to the tissue and physiopathological context, and it seems to be involved in the control of DNA synthesis, inflammatory response, and metabolism [167]. It has been recently indicated as a novel therapeutic target for cancer and is downregulated in GBM [16].

### 3.4. HDAC Inhibitors (HDACi) in GBM

Epigenetic therapy by HDACi is, to date, a well-established strategy attempting to overcome the limits of the standard treatments applied, with impressive results at both the preclinical and clinical stages, even in the presence of genetic mutations. Examples are the results obtained with valproic acid (VPA) on mouse models of recessive dystrophic epidermolysis bullosa (RDEB) [168] and the approval of Givinostat for Duchenne muscular dystrophy (DMD) [169]. With 777 clinical trials, 52 still looking for patients, their application, either alone or in combination with other drugs (e.g., kinase inhibitors, monoclonal antibodies, and standard chemotherapies) in a wide variety of cancers, is a matter of fact. The important role HDACs’ deregulation plays in glioblastoma onset and progression and the preclinical evidence of promising, beneficial effects of HDACi administration in restraining GBM growth both in vivo and in vitro provided the background required for their application also for this brain tumour. However, the results of phase I-III clinical trials employing HDACi, mainly in combination with other agents such as TMZ or Bevacizumab, were disappointing [15]. None of the HDACi tested increased the OS or PFS of enrolled patients [15]. HDACi penetration through the blood–brain barrier (BBB) may be one of the key issues in explaining the poor clinical efficacy of these epidrugs. Further, these results may also depend on the different environments and cell–cell interactions GBM cells experience in the human brain, which are extremely different from both 2D in vitro conditions and in vivo mouse brains. The isolation of patients’ GSCs, cultured as 3D neurospheres, and, more importantly, the generation of patients’ minibrain models of GBM [170] recently enriched the amount of manipulable tools that are useful for designing personalised therapies. However, whereas many works report GSC targeting by HDACi [171], no information has been acquired in GBM organoid models, which may give clues about the influence of the tumour microenvironment and extracellular matrix on HDACi efficacy. In the last 20 years, accumulating evidence provided the proof of principle that HDACs targeting may represent a doable route to fight GBM. In GBM cell lines and GSCs, it has been demonstrated that HDACi affect different pathways that are pivotal to their survival. Class I HDACs inhibitors have been shown to disrupt the chromatin superenhancers deputed to the overexpression of glycolytic genes in a Myc-dependent manner. This results in GBM metabolic reprogramming, with a shift from aerobic glycolysis to the fatty acid oxidation (FAO) metabolic pathway as a strategy to compensate for the high energy demand. Indeed, inhibiting both class I HDACs and FAO reduced cellular density, the mitotic index, and apoptosis in a GBM PDX model in vivo [172]. Similarly, the class I HDACi sodium butyrate and sodium valproate synergise with the inhibitors of glycolysis (WP1122) to reduce GBM cells’ viability [173]. HDACi also impair GSCs proliferation [174], promote GSCs differentiation, and decrease GSCs migration by limiting Wnt signalling and rearranging the cytoskeleton [175,176], impairing their self-renewal [177] also in combination with MAPK/ERK kinase (MEK) inhibitors [178] and modulating the immune microenvironment when in combination with BET inhibitors [179].

#### 3.4.1. HDACi and GBM Neoangiogenesis

Restraining GBM neoangiogenesis is another primary target for GBM treatment. Indeed, although not metastatic, GBM relies on the formation of new vessels to invade the whole brain. One of the mechanisms at the basis of GBM angiogenesis is the release of the vascular endothelial growth factor (VEGF) by GSCs and the consequent recruitment of endothelial cells (ECs) from pre-existing vessels [180]. However, anti-angiogenic therapies may not be efficient in counteracting this specific phenomenon, as GSCsc also deployed vasculogenic mimicry (VM) to self-provide a vascular tree in alternative ways compared to endothelial and smooth muscle cell transdifferentiation [181,182]. The VM network consists of tubular structures, anatomically similar to normal blood vessels but with different intrinsic properties, such as specific matrix components and location of the deposed basal lamina [183]. Given the high plasticity of GSCs, the endothelial-like derived cells begin to express true EC markers, such as CD31, VEGF receptors, and VE-cadherin to connect to host blood vessels and forming a percolative system for both nutrient and oxygen demand and the elimination of cells’ secretions [182]. These types of structures are resistant to anti-angiogenic therapies. However, it has been demonstrated that class I and II HDACi Suberoyl anilide hydroxamic acid (SAHA) and MC1568 may represent a valuable route to restrain VM in GBM [184]. Similarly, Panobinostat, a hydroxamic acid-derived pan-HDACi, reduces vessel density in GBM mouse subcutaneous xenografts and potently limits angiogenesis by disrupting HDAC6/Hsp90 interaction, promoting hypoxia-inducible factor (HIF) 1α instability and a consequent decrease in VEGF expression [185].

#### 3.4.2. HDACi and GBM Resistance to Therapies

An important property of HDACi is to sensitise GBM cells to chemo- and radiotherapy. In GSCs, Givinostat, a pan-HDACi, enhances the cytotoxic activity of TMZ in GSCs by decreasing the expression of Sp1 required for MGMT expression [186]. Panobinostat has been also shown to induce apoptosis in GBM cells in combination with TMZ and other damaging agents [187,188]. Both Trichostatin-A (TSA) and SAHA synergise with TMZ in reducing spheres formation and the proliferation of U87MG cells, increasing the levels of dual-specificity phosphatase 1 (DUSP1), a MAP kinase phosphatase that positively correlates with patient survival. Indeed, high DUSP1 levels are correlated with increased GBM cell differentiation and reduced tumorigenic ability, both in vitro and in vivo [189]. SAHA also synergises with melatonin in reducing the levels of transcription factor EB (TFEB), which is a master regulator of autophagy and lysosomal biogenesis and is overexpressed in high-grade GBM, and promoting apoptotic gene expression. DNA damage, as demonstrated by the occurrence of DSBs in GBM cells treated with the combination of SAHA and melatonin, and PARP cleavage are part of the mechanism underlying SAHA/melatonin-induced apoptotic GBM cell death [190]. However, SAHA has been also associated with TMZ resistance through MGMT upregulation via a chromatin-based mechanism, including NF-κB, Sp1, Jun, and p300 recruitment at its own promoter in GBM xenograft models [191]. Class I and II HDACi Romidepsin potentiates the effects of TMZ, inducing apoptosis of GBM cells by increasing the expression of pro-apoptotic genes and restraining the PI3K/Akt pathway both in vitro and in vivo [192]. The HDAC8 inhibitor NBM-BMX has been reported to counteract TMZ resistance in GBM cell lines by reducing the level of MGMT and increasing the expression of p53. A decrease in c-Myc and Cyclin D1 levels was also observed. At the biological level, NBM-BMX, in combination with TMZ, inhibits cell viability and proliferation and promotes apoptosis [193]. The HDAC1 and 3 inhibitor RGFP109 alters the binding capacity of NF-κB, blocking the association to its co-activators (P/CAF and p300) and promoting the interaction with the inhibitor of growth 4 (ING4), downregulating the expression of pro-survival genes [194].

#### 3.4.3. HDACi Crossing the BBB

Except for MS-275, a potent class I HDACi [195] and givinostat [177], as stated above, hitherto, the presence of the BBB represents an obstacle to the efficient application of pre-clinically validated epidrugs in humans. Thus, a major effort is ongoing in the design of new molecules capable of penetrating the BBB and reaching the brain at a clinically valuable concentration. Quisinostat, a hydroxamic acid derivative, has been shown to be cytotoxic, to promote the accumulation of DSBs in GSCs, and thus to radiosensitise GSCs in vitro, and to cross the BBB, reducing the tumour mass in vivo [196]. Largazole, a natural, orally bioavailable, class I HDACi from marine cyanobacteria, also crosses the BBB, leading to changes in the gene expression pattern of GBM-affected mouse brains, where the genes involved in neuroprotection become upregulated [197]. Other hydroxamic acid derivatives have been developed to allow BBB crossing. SAHA analogues, with different caps, have been proven to be effective in reducing U87MG cell viability while inducing reactive oxygen species (ROS) accumulation and apoptosis [198]. Further, these molecules have been shown to inhibit HDAC8 activity in GBM and HDAC2 and 3 expression. Importantly, these SAHA analogues downregulate the expression of mTOR signalling components, such as Rictor, whereas p53 and mir143, which targets hexokinase2 (HK2), the first enzyme involved in glycolysis and overexpressed in GBM [199], are upregulated [198]. Panobinostat, when encapsulated into nano-micelles, may cross the BBB in rat models and has anti-proliferative effects in a variety of GBM cell lines in vitro [200]. Piperazine-based HDACi benzamides have been also shown to cross the BBB in mouse models and to potently impair GBM cell lines proliferation by affecting cell cycle-related pathways, such as p16INK4a-CDK4/6-pRb [201].

## 4. HDAC6

The HDAC6 coding gene is located on chromosome X p.11.22-23 and is about 22 kb long. As a member of class IIb HDACs, the 1215 amino acids corresponding gene product is provided with two catalytic domains (CD1 and 2) and a Zn^2+/^/ubiquitin binding pocket (ZnF-UBP; Figure 2), which makes this deacetylating enzyme unique within the HDACs family. One NLS and one NES (NES1) are located at the N-terminus of the molecule, while NES2 is at the C-terminus. Between the two CDs, a dynein motor proteins binding domain (DMB) is found, whereas a SE14 tetra decapeptide repeat (for a total of eight tetra decapeptides) is located between the second catalytic domain and NES2. This region, which is proper for the sole human HDAC6, is required for cytoplasmic retention [156]. CD1 requires CD2 for its activity, and both require Zn^2+^ to exert their function. The DMB motif provides an anchor for microtubules and the related transport of different cargo along microtubules [202].

Different post-translational modifications affect HDAC6 activity. Indeed, p300-dependent acetylation negatively regulates HDAC6, whereas ubiquitination may either not affect [203] or lead to HDAC6 degradation [204,205]. Some residues, when phosphorylated, increase HDAC6 activity, (S1060 targeted by GPCR kinase 2 (GRK2) [205]; T1031, S1035 phosphorylated by extracellular signal-regulated kinase (ERK) [206]; and S22 phosphorylated by glycogen synthase kinase 3β (GSK-3β) [207], while others are detrimental, e.g., Y570, which is a target of epidermal growth factor receptor (EGFR) [208]. Additional HDAC6 activating kinases are Aurora A kinase (AurA) [209], p38α, and protein kinase Cα [210]. Furthermore, HDAC6 has been also found to be S-nitrosylated and inactive in this specific form [211].

### 4.1. HDAC6 Functions

Located primarily in the cytoplasm, HDAC6 exerts its activity—which includes both the deacetylation, binding, and aggregation of ubiquitinated proteins and the ubiquitination of certain targets, mainly on non-histone targets. However, HDADC6 plays an important role also in the cell nucleus.

#### 4.1.1. Deacetylation-Dependent Functions of Nuclear HDAC6

In cycling cells, HDAC6 is preferentially cytoplasmic. However, in resting cells, a fraction of HDAC6 translocates to the nucleus which participates in the regulation of gene expression by binding to and regulating the activity of transcription factors (TFs) and transcriptional co-repressors/activators. Examples of TFs that recruit HDAC6 into the nucleus are Runx2 and NF-κB [212,213]. These proteins take advantage of HDAC6 to repress genes in a tissue-specific manner. HDAC6-mediated transcriptional repression usually foresees the involvement of co-repressors (e.g., HDAC11 and ligand-dependent nuclear receptor co-repressor (LCoR)) [214] or activators that are modified in order to lose the ability to induce transcription. Indeed, when sumoylated, p300 acts as a transcriptional repressor by HDAC6 recruitment [215,216,217]. On the other hand, p300 acetylates HDAC6, impairing its tubulin-deacetylating activity [218] and promoting its cytosolic retention by blocking its interaction with importin-α [219]. HDAC6 also controls p53 activity. Indeed, it is responsible for p53 deacetylation at K381/382, strongly limiting the p53-dependent transcription of growth arrest and apoptotic genes [220]. HDAC6 exerts this specific function within the nucleus, as changes in the expression levels of genes controlled by p53 are dependent upon the nuclear HDAC6 amount [220].

Importantly, HDAC6 has been found to deacetylate and regulate the proteins belonging to the DNA mismatch repair system (MMR), including MSH2 [221] and MLH1. This latter, when deacetylated, impairs the formation of the MutSα–MutLα complex, thereby inducing DNA damage tolerance [222]. This specific HDAC6 activity is particularly important in promoting the chemo- and radioresistance of cancer cells (see below).

The nuclear quota of HDAC6 may also deacetylate histones [223,224,225,226]. In addition, indirect mechanisms may account for HDAC6 histone deacetylase activity, such as the cytosolic sequestration and the ubiquitination of HATs [227].

#### 4.1.2. Deacetylation-Dependent Functions of Cytosolic HDAC6

It is well established that HDAC6 exerts its activity mainly on cytosolic proteins, being its primary substrate α-tubulin, the principal component of microtubules. α-tubulin is acetylated by α-tubulin acetyltransferase 1 (αTAT1) [228]. The first demonstration of HDAC6-dependent α-tubulin K40 deacetylation dates from 2002 [229], followed by the identification of cortactin, which regulates actin polymerisation and branching through the binding to F-actin, as another HDAC6 substrate [230]. Deacetylation of α-tubulin leads to microtubule depolymerisation and improved cell motility. This activity is enhanced when HDAC6 is phosphorylated onto S1035 by the extracellular regulated kinase 1 (ERK1). In addition, HDAC6 alters the binding capacity of cortactin to F-actin [230], thus affecting not only the microtubule dynamics but also actin-based cell motility. A paradigm of the importance of this specific activity of HDAC6, associated with the presence of a DMB, is represented by its role in both the central and peripheral nervous systems, as axons rely on the transport of molecules and organelles (i.e., mitochondria) along cytoskeletal components for their survival. Specifically, it has been shown that HDAC6 may affect axonal transport deacetylating α-tubulin. Indeed, acetylated α−tubulin provides a docking site for motor proteins, and it has been hypothesised that the dynamic acetylation/deacetylation of α-tubulin may represent a code for specific motor proteins [231]. Kinesins, for example, which are responsible for the anterograde transport along microtubules, preferentially recognise acetylated α-tubulin [232]. Furthermore, HDAC6 may also influence mitochondria axonal transport. In fact, HDAC6 deacetylates Miro1, a calcium-binding outer mitochondrial membrane protein, which, when acetylated on K105, increases mitochondria transport along axons [233,234]. Thus, HDAC6 negatively affects axonal transport by α-tubulin and Miro1 deacetylation and, consequently, impairs axonal outgrowth. The latter is also dependent upon the cortactin–F-actin network [235] at the growth cone. As mentioned before, acetylated cortactin drives F-actin polymerisation [230], which is required for the membrane protrusion that, in turn, fuels the leading edge and axon outgrowth. Therefore, cortactin deacetylation is another mechanism through which HDAC6 severely impairs axonal outgrowth [230,236].

In addition to its role in cytoskeleton dynamics and cargo transport along microtubules, HDAC6 deacetylates also a number of signalling molecules. Indeed, HDAC6 binds and deacetylates ERK 1 and 2, enhancing their activity [237], while, as mentioned above, ERK1 phosphorylates HDAC6, resulting in a self-fuelling circuit that controls both the response to extracellular signals (e.g., growth factors) and, consequently, cell proliferation and motility [206]. In addition, HDAC6 participates in both the PI3K/Akt and the Wnt signalling pathways. By deacetylating HSP90, HDAC6 reinforces Akt/HSP90 binding, providing protection for HSP90 from proteasomal degradation and stabilising Akt phosphorylation [156]. By deacetylating β-catenin, HDAC6 allows its nuclear translocation and activation of Wnt-dependent transcription [238].

#### 4.1.3. ZnF/UBP-Dependent HDCA6 Activity

The ZnF/UBP binding domain of HDAC6 virtually hooks every ubiquitinated protein due to its high affinity for ubiquitin (ub)-residues. This property makes HDAC6 actively involved in the elimination of misfolded protein aggregates through the autophagy pathway when the proteasome is inhibited [239,240]. Indeed, when the proteasome machinery is functional, HDAC6 interacts with valosin containing protein (VCP)/p97. This interaction promotes the dissociation of HDAC6 from the ubiquitinated target and the degradation of the latter through the proteasome [241]. However, when the proteasome is inhibited, HDAC6 moves misfolded aggregates to the perinuclear region at the microtubule organising centre (MTOC) through dynein, which is responsible for retrograde transport in the microtubules, contributing to aggresomes formation, which are further degraded by the autophagy pathway [242]. HDAC6 also impairs the formation of misfolded protein aggregates through the activation of HSPs. Indeed, in physiological conditions, HDAC6 is bound and deacetylates HSP90/heat shock factor 1 (HSF1). When polyubiquitinated proteins are bound by the ZnF/UBP domain of HDAC6, HDAC6 detaches from and releases HSP90, which becomes acetylated, and HSF1, which induces the transcription of HSP70 and 27. These chaperones, together with HSP90, impair the formation of new misfolded aggregates [240].

Very recently, it has been also demonstrated that HDAC6 participates in aggresomes’ structure maintenance by interacting with vimentin p72 and locking the aggresome vimentin cage [243], uncovering another intriguing function for HDAC6.

In addition to aggresomes, the ubiquitin binding and motor activities of HDAC6 control also the formation and activation of the NACHT, leucine-rich repeat (LRR), and pyrin domain (PYD)-containing protein 3 (NLRP3) inflammasomes, which protect the cell from endogenous dangerous signals (due to both cell damage and death and a variety of pathogens) employing pro- and activated caspase 1 and pyroptosis and inducing the secretion of interleukins (1 and 18) [244,245]. HDAC6 binds to NLRP3, mediates its reverse transport to the MTOC [242], and activates the NLRP3 inflammasome, a process dependent on the HDAC6 ubiquitin-binding domain. Moreover, HDAC6 interacts with ubiquitinated NLRP3, promoting its degradation in an autophagic manner [246]. In parallel, the prominent affinity for ubiquitin residues makes HDAC6 capable of regulating viral infections, especially of influenza A (IAV), human immunodeficiency 1 (HIV-1), and Coxsackie A16 viruses (CA16). In this regard, HDAC6 exerts both pro-viral and antiviral activities. Indeed, by binding the M1 tag on the IAV capsid, HDAC6 transports IAV particles along the microtubules. After M1 removal and virus uncoating, caspase-3-dependent degradation of the ZnF-UBP domain of HDAC6 allows for viral replication and virion assembly [247]. At the same time, stabilisation of the HDAC6 SE14 domain induces cell apoptosis, spreading the virus outside the dying cell [248]. However, HDAC6 may also block IAV replication and spread by directly inhibiting PA ribonuclease and RNA polymerase and reducing the transport of virions along microtubules by deacetylating α-tubulin [249,250].

HDAC6 is also a “guardian” that monitors infections by HIV-1. Indeed, by binding to and moving Vif and Pr55Gag, two key players in HIV-1 assembly and release, to the autophagosomes, HDAC6 restricts HIV-1 infection [251]. On the other hand, the HIV-1 core helper protein Nef promotes HDAC6 degradation, ensuring HIV-1 dissemination [252].

CA16 is a common enterovirus, causing hand–foot–mouth disease. The PKR/eIF2α pathway is exploited by the cell to form stress granules (SGs) [253,254] that limit viral replication and induce the production of antiviral molecules, such as interferons. The ZnF/UBP domain of HDAC6, together with p62, promotes SGs degradation, inhibiting type I interferon release and allowing viral replication [255].

#### 4.1.4. Ubiquitinating Activity

In addition to binding to ub-residues, HDAC6 has been found to possess a ubiquitin E3 ligase activity. Indeed, HDAC6 ubiquitinates DNA repair enzymes belonging to both the base excision repair (BER) and the mismatch repair systems (MMR). It has been found to ubiquitinate MSH2 [221], Chk1 [256], and p300 [227].

### 4.2. HDAC6 and Cilia

Both the critical activity of HDAC6 in regulating the assembly of cytoskeletal components (i.e., microtubules and actin fibres) and the ZnF/UBP-dependent functions are linked to the important role this enzyme plays in primary cilium formation/resorption (Figure 1). Although the role of HDAC6 in primary cilia disassembly and signalling is still an active area of investigation, it has been uncovered that the AurA kinase-mediated phosphorylation of HDAC6 results in its activation, axoneme’s microtubules deacetylation, and cilia retraction [209]. During this process, HDAC6 is assisted by a wide number of mediators, e.g., human enhancer of filamentation 1, pitchfork, calmodulin, nephrocystin-2/inversin, macro-phage stimulating 1/2 (Mst1/2), and β-catenin, which regulate HDAC6/AurA interaction [257,258,259,260,261]. In parallel, HDAC6 is differentially regulated by the proteins involved in cilia formation and signalling, being either inhibited [262] or activated [263] or intracellularly re-localised [264]. Further, stress signal-dependent cilia shortening relies on the ability of HDAC6 to eliminate the aggregates of misfolded intracellular ciliary proteins that form when cilia are damaged by toxic agents, such as cigarette smoke. In this process, misfolded aggregates are recognised by HDAC6 and moved to the autophagosome in order to be transferred to lysosomes for autophagic degradation [265].

### 4.3. HDAC6 in GBM

The role of HDAC6 in cancer has been extensively reviewed elsewhere [156,266]. In this paragraph, we will strictly focus on the functions HDAC6 performs in the onset and progression of GBM. It has to be remarked that, to date, very few papers investigated the role of HDAC6 in glioblastoma ontogeny. In addition, most of the findings have been similarly reported in both GBM cell lines and patient-derived GSCs. Thus, in this paragraph, a summary of the whole amount of available data will be provided, summarised, and schemed in Figure 3.

Despite the little piece of information, HDAC6 has been found to be involved in pathways pivotal to GBM cell survival, such as proliferation, spheroid formation, migration, de-differentiation, and resistance to both radio- and chemotherapy, due to its involvement in DNA damage tolerance.

#### 4.3.1. HDAC6 and GBM Cells’ Proliferation

It is widely accepted and demonstrated that GBM cells, including GSCs, exhibit high levels of HDAC6. HDAC6 promotes the proliferation of GBM cells in a number of ways, a phenomenon that is classically paralleled by radio- and chemotherapy resistance. First, by potentiating ERK1 and 2 activity, it strongly amplifies EGF receptor mitogenic signalling and potentiates GBM spheroids formation. Further, stabilising EGFR also promotes resistance to TMZ [21]. By interacting with carbon catabolite repression, negative on TATA-less (CCR4-NOT) core exoribonuclease subunit 6 (CNOT6), a typical protein related to mRNA decay, and FUS RNA protecting protein, HDAC6 protects the long non-coding RNA LINC00461 by preserving polyadenylate tail maintenance. This HDAC6-dependent mechanism results in an increase in the proteins related to cell division by a sponging mechanism through which LINC00461 sequesters miR-485-3p [267]. HDAC6 has been also found to increase the activity of TFs deputed to the regulation of cell cycle-related genes. Specifically, HDAC6 increases MAP kinase 7 (MKK7) protein levels and downstream Jun N-terminal kinase (JNK)/c-Jun activities, bursting GBM cell proliferation. In this regard, HDAC6 protects MKK7 from proteasomal degradation [22]. This mechanism of protection could rely on MKK7 deacetylation by HDAC6, as occurs for Flotillin-2 (FLOT2) in nasopharyngeal carcinoma [268]. In addition, in cooperation with HDAC1 and 2, it induces Sp1 transcription by a positive feedback loop in which Sp1 deacetylation is required for its binding to its own promoter. Consequently, enhanced Sp1 activity has been shown to upregulate cell cycle genes, promoting GBM cell proliferation [269]. Gli1 is also controlled by HDAC6 at the transcriptional level. Gli TFs are downstream players of the Shh pathway, which is involved in GBM cell proliferation and GSC stemness maintenance (see also section “Altered Primary Cilia Siganlling in GBM” [73,270,271]). Indeed, it has been found that, upon HDAC6 silencing or chemical inhibition by tubastatin A (see below), Gli1 mRNA decreases in GBM cell lines and that this regulation is transcription-dependent [28,272]. Thus, sustaining Gli1 activity, HDAC6 exploits Shh signalling to propel GBM cell proliferation. Smad2 is another TF affected by HDAC6. Indeed, this latter inhibits Smad2 phosphorylation, impairing its binding to the p21 promoter and inducing GBM cell proliferation [23].

#### 4.3.2. HDAC6 and the Epithelial–Mesenchymal Transition (EMT)

Very little information is available regarding the role HDAC6 plays in GBM EMT. It has been shown that HDAC6 regulates EMT TFs Snail and Slug expression, presumably at the transcriptional level (or via increased mRNA stability), although a molecular mechanism has not been provided [28]. However, given the role that HDAC6 exerts in this process in other cancers [273,274,275], it is conceivable that GBM spreading and invasion of the surrounding brain tissue would, at least in part, depend on HDAC6 activity.

#### 4.3.3. HDAC6 Action Through GSCs Primary Cilia

Very few reports address the role of HDAC6 in controlling pivotal GSC properties through the modulation of primary cilia assembly/disassembly and signalling. Inhibiting HDAC6 enzymatic activity produces an increase in both the acetylated levels of α-tubulin and in the number of ciliated, patient-derived GSCs, as expected. Intriguingly, in the same experimental conditions, α-tubulin within the cilia axoneme is deacetylated. As a consequence, cilia depletion through KIF3A and ARL13B knockdown does not entail GSC growth arrest upon HDAC6 inhibition, but rather a block of differentiation. Thus, in GSCs, HDAC6 acts through the primary cilia to restrain differentiation programs (see also Section 2.2.2), maintaining their proliferation ability [19]. This phenomenon may be dependent on overactive Shh signalling in HDAC6-overexpressing GSCs. Indeed, in HDAC6-depleted cells, Gli TFs and patched receptors are downregulated and the Shh pathway is inactive, leading to GSC differentiation [18].

#### 4.3.4. HDAC6 and GBM Resistance to Radio- and Chemotherapy

Given the established role of HDAC6 in activating diverse DNA damage response (DDR) pathways [222,276,277,278], its role in mediating both radio- and chemotherapy resistance in GBM is not surprising. HDAC6 is known to activate multiple DDR genes via Sp1 [20], thus promoting TMZ resistance in GBM cells. It has been reported that, together with HDAC4, HDAC6 promotes double-strand breaks (DSBs) repair in glioblastoma, as interfering with their translation promotes radiosensitivity in the U87MG and U251 GBM cell lines but not in normal astrocytes or cortical neurons. HDAC6 interference not only provided the accumulation of γH2AX spots onto chromatin but also impaired the nuclear translocation of ATM and DNA-Pkcs, two key players involved in DSBs repair, which interact with HDAC6. HDAC6 silencing induces both apoptosis and autophagy in GBM cells and, more importantly, negatively affects GSC’s self-renewal [24]. HDAC6 inhibition also promotes the accumulation of MSH2 and 6, as well as the decrease in MGMT and p53, inducing apoptosis in TMZ-resistant GBM cells [273]. In GSCs, HDAC6 inhibition has been shown to promote Chk1 degradation through the downregulation of the X-linked inhibitor of apoptosis (XIAP), thus promoting radiosensitivity [26].

#### 4.3.5. GBM Subtypes: A HDAC6 Point of View

Although no information is available about the selective functions of HDAC6 in the three different GBM subtypes, some insights may be deduced by the HDAC6-dependent regulation of key players identifying a specific GBM subtype. The involvement of HDAC6 in EGFR and Shh signalling amplification [21,73,270,271] suggests a role in the onset and/or maintenance of the classical GBM subtype. In parallel, HDAC6’s role in promoting neoangiogenesis and in regulating inflammation and cell migration indicates that deregulation of HDAC6 activity may also influence pathway derangement in mesenchymal GBM cells [4,212,213]. It has been shown that HDAC6 controls VEGF-dependent notch signalling, promoting angiogenesis in lung ECs [279]. Further, by Sp1 deacetylation and the consequent increased transcriptional activity, HDAC6 regulates endoglin, a co-receptor activating transforming growth factor-β (TGF-β) signal transduction, expression and tumour angiogenesis [280,281]. Strikingly, HDAC6-dependent cortactin deacetylation (see Section 4.1.2) initiates ECs migration [281].

The role HDAC6 plays in regulating cilia disassembly directly impacts the tumorigenic properties of the proneural subtype cells, which are characterised by alterations in PDFGRα (focal amplification and mutations [2]). By PDGFRα de-localisation from the ciliary membrane to the cell cytosol, due to cilia disassembly, HDAC6 may contribute to PDFGRα signalling deregulation and proneural GBM cell maintenance [104].

### 4.4. HDAC6 Inhibitors

FDA-approved HDACi for clinical use has the disadvantage of causing consistent side effects when administered to humans [15]. Indeed, vorinostat, panobinostat, belinostat, romidepsin, and chidamide [282,283,284,285] are pan-HDACi that impact systemically on diverse organs, provoking asthenia, cardiotoxicity, diarrhoea, fatigue, and a severe decrease in blood cell populations [286,287,288,289]. In contrast, selective HDACi may be more well-tolerated, with mitigated side effects. Over the years, different specific HDAC6i have been developed, taking into account the “gold” of the pharmacophore model common to all HDACs, namely the cap, the linker, and the zinc-binding group (ZBG). In general, the cap is an aromatic group connected to the hydrophobic alkyl group representing the linker and the ZBG coordinates the Zn^2+^ ion in the active site. HDAC6i produced are discriminated by the ZBG, which may be a hydroxamic derivative or another ZBG. Among the hydroxamic acid-based HDAC6i, Tubacin was the first to be discovered [290,291] showing an almost 320-fold selectivity over HDAC1 and 2. Tubacin has no severe side effects, but it cannot be used for clinical purposes, as it is highly lipophilic. Based on the tubacin structure, Ricolinostat has been proven to be a better drug. It is the first HDAC6i approved for human use to treat multiple myeloma (MM) [292] and synergises with both bortezominb and carfilzomib (proteasome inhibitors) to induce apoptosis of MM cells [293]. Linear triazolylphenyl–hydroxamic acid derivatives have been shown to be selective HDAC6i with NOC-7 being the most potent (IC_50_ = 0.002 nM) [294]. Also, biaryl hydroxamates possess an nM inhibitor activity over HDAC6 [295]. SAHA (vorinostat) derivatives, with modifications of C_2_, C_4_, and C_5_ in the linker, also showed a certain selectivity for HDAC6 [296].

Considering the wider and almost outward channel of HDAC6, new inhibitors have been developed that replace the alkyl chain linker with aromatic groups and are designed to have larger and stiffer caps, which were supposed to be more effective in inhibiting HDAC6 activity. Tubastatin A, a tetrahydro-γ-carboline derivative, is one of these inhibitors and shows a 1000-fold selectivity for HDAC6 over all of the HDACs, except HDAC8 (57 fold) [297]. Intriguingly, Tubastatin A not only promotes α-tubulin hyperacetylation but also histone acetylation, thus also suggesting an impairment of HDAC6 nuclear function. Second-generation and sulphur analogues of Tubastatin have been developed, showing a ~5000–7000 fold selectivity over class I HDACs and a subnanomolar inhibitory activity over HDAC6 [298,299]

“Y-shaped” HDAC6i also hold potent inhibitor activity over HDAC6 at nanomolar concentrations. Nexurastat is 601-fold more selective for HDAC6 over HDAC8 and possesses a branching element near the ZBG, which accounts for its specificity [300]. Similarly, HPOB is selective for HDAC6, with an IC_50_ 10-fold higher than that of Nexurastat [301].

Many other HDAC6i have been designed, although their efficacies were proven at submicromolar concentrations, and they frequently inhibited not only HDAC6 but also HDAC8 [302,303]. Of note, hydroxamates are genotoxic. Therefore, the recent tendency is to provide HDAC6i with other ZBGs, such as mercaptoacetamide, thicarbonate, trifluoromethylketone, hydrazide, hydroxypyridones, and others. However, these molecules still deserve further improvements [302].

#### 4.4.1. HDAC6 PROTACS

An intriguing novel class of HDAC6i is represented by proteolysis-targeting chimaeras (PROTACs), which are bifunctional molecules provided with a small molecule inhibitor and a ligand of the ubiquitin E3 ligase [304], which promote proteasome degradation of the protein of interest. This technology fully revolutionised the field of small molecules based exclusively on the chemical inhibition of the catalytic pocket, as they provide the complete elimination of the protein of interest (POI). Further, they showed high versatility with respect to small molecules targeting POI active sites, as they may be transiently linked to any fissure present in their target. First reported at the beginning of the XXI century, PROTACs research has skyrocketed in the past 5 years and is applied to different POIs, including TFs, chromatin remodelling enzymes, and nuclear and membrane receptors, among others [305]. Importantly, two entered clinical trials, namely ARV-110 (clinical trial identifier NCT NCT03888612) and ARV-471 (clinical trial identifier NCT04072952), targeting the androgen and the estrogen receptor, respectively. PROTACs present several advantages compared to small molecule inhibitors, including isoform selectivity, disruption of enzymatic and scaffolding activities, and degradation of members of multimolecular complexes [305]. Different PROTACs having HDAC6 as a POI have been developed. Hydroxamate-based, cereblon-based and PROTACs designed onto Y-shaped HDAC6s [306,307,308] have been demonstrated to potently degrade HDAC6 and to be efficient in restraining HDAC6-dependent inflammatory events and impair cancer cells’ proliferation [309,310,311].

#### 4.4.2. Dual Inhibitors

A characteristic of most of the HDACis is that their efficacy is basically limited to haematologic malignancies. Thus, dual inhibitors have been designed to be applied also to solid cancers. 6-(4-((5-Chloro-4-((4-chlorophenyl) amino) pyrimidin-2-yl) amino)-1H-pyrazol-1-yl)-N-hydroxyhexanamide is a JAK/HDAC inhibitor. When the 4-chlorophenyl moiety is substituted by pyridine or other phenyl groups, the selectivity for HDAC6 increases, together with the anti-tumour properties [312]. Derivatives of VEGFR and HDAC inhibitors were also produced and showed anticancer activity [313]. GSK3β/HDAC6 dual inhibitors and HDAC6/tubulin inhibitors have been also developed and showed significant efficacy both as anticancer agents and neuroprotective drugs [314,315].

#### 4.4.3. HDAC6i in Clinical Trials

To date, eight clinical trials employing HDAC6 inhibitors, alone and in combination with other drugs, for cancer treatment have been launched. Half were completed. Two stopped early. One was withdrawn, and one is still recruiting. The results from the completed trials are encouraging, as the therapies were safe and well-tolerated. Further, responses, although partial, were observed. Ricolinostat (ACY1215) in combination with bortezomib, a proteasome inhibitor, and dexamethasone has been proven to be relatively efficient in relapsed MM patients (37%) and, most importantly, in a small percentage of bortezomib refractory patients (14%) [292]. Citarinostat (ACY241), either in combination with nivolumab or paclitaxel, has been tested in patients affected by solid tumours. In 17 non-small cell lung cancer (NSCLC) patients, no dose-limiting toxicity was reported, and either partial response or stable disease was achieved with increased infiltrating CD3^+^ lymphocytes [316]. The same promising results were obtained in 20 patients affected by non-hematologic malignancies. No dose-limiting toxicity, partial responses, and stable diseases were observed together with decreased paclitaxel-induced neutropenia [317]. KA2507 (IC_50_ = 2.5 nM) similarly showed no dose-limiting toxicity in patients with PD-L1-expressing tumours. The best response detected was a stable disease in patients with relapsed and/or refractory solid tumours [318]. Overall, these results strongly suggest the use of HDAC6i, alone or in combination, as a powerful strategy to treat advanced and refractory cancers. Moreover, the lack of considerable side effects, which are, instead, provided by pan-HDACi, makes HDAC6i a more advisable strategy for cancer treatment.

### 4.5. HDAC6i Application in GBM

HDAC6i application for GBM therapy is still an unexplored field of investigation. Few works have reported the use of HDAC6i in GBM models. However, the available literature is unanimous in suggesting HDAC6 inhibition as a valuable tool to counteract GBM progression and, more importantly, resistance to therapy. A variety of mechanisms and biological effects are at the basis of HDAC6i efficacy in both GBM cells and GSCs and in animal models. Tubastatin A has been demonstrated to inhibit migration, the Shh pathway, and to accelerate apoptosis in TMZ-treated GBM cell lines [272]. Dual inhibition of Shh and HDAC6 by tubastatin A and cyclopamine significantly inhibits GBM cells’ proliferation in vitro and in orthotopic transplants in zebrafish hindbrain ventricles. Mechanistically, the combo treatment results in lysosomal stress with an impaired fusion of lysosomes and autophagosomes and derailment of sphingolipids catabolism in GBM cell lines and zebrafish embryos [27]. Additional evidence has been provided regarding the impairment of the autophagic pathway by HDAC6i as a route to induce GBM cell apoptosis. Indeed, tubastatin A and TMZ reverse the endoplasmic reticulum stress tolerance (ERST), a phenomenon that accounts, at least in part, for TMZ resistance and induces pro-apoptotic signals by activating the unfolded protein system (UPS) and harmful ER stress. Tubastatin A alone reduced the HDAC6 levels in autophagosomes of GBM cells, whereas the combination treatment promotes the association of HDAC6 to VCP/p97, resulting in defects in the clearance of misfolded proteins through the HDAC6-dependent autophagic pathway while promoting UPS [319]. Similarly, tubacin promotes autophagosome accumulation, growth arrest, and death in patient-derived glioma cells. In combination with TMZ, these effects were greatly enhanced, suggesting that fusion impairment of autophagosomes with lysosomes could be a common mechanism of action of HDAC6i [320]. In TMZ-resistant GBM cells, tubastatin A and different HDAC6i induce MMR genes and promote apoptosis. Additionally, a decrease in EGFR phosphorylation is observed. HDAC6i also induce wild-type p53 in TMZ-sensitive patients and decrease mutant p53 expression in TMZ-resistant GBM cells [25].

HDAC6 PROTACS have been also proven to be a useful strategy to inhibit GBM growth. The J22352 compound promotes the degradation of aberrantly expressed HDAC6 in GBM, with the consequent autophagic-dependent cell death, block of cell migration, and tumour growth inhibition in mouse subcutaneous xenografts. More importantly, J22352 reduced the level of immunosuppressive PD-L1 while dramatically recruiting CD8^+^T lymphocytes at the tumour site in vivo [321].

Dual inhibitors have been also designed and tested as efficient in the reduction of GBM growth. Adducts of donepezil, a drug approved for Alzheimer’s disease, and different linkers and hydroxamate ZBGs have been produced. One of them, with an IC_50_ of 2.7nM for HDAC6 and 710nM for HDAC2, showed the greatest efficacy in limiting GBM cell growth in vitro and in vivo and in overcoming TMZ resistance [322]. A dual inhibitor of cytochrome P450 17A1 (CYP17A1), which is correlated with a poor prognosis in GBM patients and resistance to TMZ [323], and HDAC6 has been also produced. This abiraterone [324] (CYP17A1 inhibitor)/hydroxamic acid inhibitor also showed a potent antiproliferative effect on GBM both in vitro and in vivo, enhanced ROS production, activated DDR, and reversed TMZ resistance [325].

## 5. Conclusions

As outlined in the present work, HDAC6 appears as a multifaceted molecule ruling a plethora of biological mechanisms, spanning through the regulation of cytoskeleton-dependent processes until transcriptional regulation [156]. As a consequence, its inhibition has to be carefully evaluated when considering HDAC6 inhibition to treat human disorders. However, several pieces of evidence effectively suggest HDAC6 targeting as a route to face different diseases.

### 5.1. From NDs to Cancer: Blocking Deacetylating and ZnF-UBP-Related Functions of HDAC6

The fundamental role HDAC6 plays in controlling the polymerisation/depolymerisation of α-tubulin and actin implies a major involvement in CNS and PNS proper functionality. Defects in axonal transport are typical of neurodegenerative disorders (NDs), as well as the presence of aggregates of misfolded proteins and neuroinflammation due to inflammasome activation [326,327,328]. The involvement of HDAC6 in the axonal transport and regulation of misfolded aggregates and inflammasomes links HDAC6 to NDs, such as Alzheimer’s disease (AD), Parkinson’s Disease (PD) [328], and peripheral neuropathies such as Charcot Marie Tooth (CMT) syndrome [329] and chemotherapy-induced peripheral neuropathy [330]. Further, HDAC6 inhibition has been shown to reverse transactive response DNA-binding protein-43 (TDP-43)-related axonal defects [331,332], suggesting HDAC6 targeting as a still-unexplored treatment for NDs.

Similarly, its role in regulating primary cilia disassembly may envisage HDAC6 as a molecular target for the treatment of ciliopathies, a family of diseases presenting both genotypic and phenotypic heterogeneity, primarily affecting the CNS [83,84]. In this context, HDAC6 functions are not yet fully investigated, but it is highly probable that, in addition to its role in controlling microtubule depolymerisation, HDAC6-dependent deacetylation of ciliary proteins or adaptors/mediators may participate in both the structural and signalling complexity of this cellular antenna. Different HDAC6 functions may be clinically targeted in an attempt to counteract primary cilia dysfunction. Inhibiting its deacetylation activity against α-tubulin could be the primary objective, as this should result in cilia restoration [209,257,258,259,260,261]. Also, the inhibition of HDAC6 ZnF-UBP-dependent autophagic activity could represent a potential goal to restrain cilia shortening [265]. Although a link between cilia and autophagy has been clearly established, we are still at the dawn of this emerging field of investigation, and further studies are necessary to dissect this complex circuitry [332]. Indeed, studies reporting the role of the autophagic process in ciliogenesis have been contradictory. It has been observed that autophagic degradation of the oral–facial–digital type 1 (OFD1) protein promotes ciliogenesis [333], whereas an autophagy-dependent depletion of ITF20 causes the opposite [334].

The involvement in viral infections (see Section 4.1.3) indicates HDAC6 as another potential weapon to prevent and/or treat viral diseases. With regard to inflammation-related disorders, it has to be noted that, depending on the specific context, HDAC6’s role in regulating the inflammasome may be beneficial or detrimental. Thus, while in NDs blocking inflammation and HDAC6-dependent inflammasome activation may be exploited as a strategy to prevent, treat, and/or ameliorate NDs symptoms [326], in cancer, the inhibition of this specific HDAC6 function is a double-edged sword. Indeed, chronic inflammation activates the NLRP3 inflammasome, which triggers an immunosuppressive response, tumour progression, and metastasis formation [335]. However, the opposite has been also observed. In colorectal cancer, NLRP3 inflammasome activation is associated with natural killer cells recruitment and tumour suppression [336]. Furthermore, inflammation-associated IL-18 production has been proven to possess tumoricidal activity [337]. Thus, the efficacy of the inhibition of inflammasome-associated HDAC6 functions is highly context-dependent.

### 5.2. HDAC6 Inhibition: A Novel Strategy Against GBM?

HDAC inhibitor application for GBM treatment surprisingly results in failure when administered mainly as a combo therapy in clinical trials. Multiple reasons may account for HDACi’s inefficacy in GBM. One of major importance is the presence of the BBB, which preserves the brain from insults. In addition, tumour heterogeneity and the surrounding microenvironment may provide a further level of complexity, which may strongly limit HDACi’s success. Over the years, the drug discovery field moved to the design of novel hybrid molecules able to cross the BBB [200,201] and to simultaneously target HDACs and other players pivotal to glioblastomagenesis [322,338,339]. However, thorough preclinical investigations are required to determine the effectiveness of these molecules.

Despite the paucity of information, inhibiting HDAC6 is emerging as a potential novel tool to enrich the extremely poor arsenal of treatments available against GBM. Indeed, high HDAC6 levels are constantly associated with (i) enhanced GBM cell proliferation and migration; (ii) increased resistance to radio- and chemotherapy; and (iii) self-renewal of GSCs. The amplification of Shh signalling and the resorption of primary cilia are characteristic of both HDAC6 activity [73,270,271] and GSCs, further strengthening the importance of GSCs targeting to the treatment of GBM. In this regard, resistance to current therapies is a specific trait of GSCs, capable of tolerating high levels of DNA damage [340]. Notably, HDAC6 has been found to interact with a plethora of DNA repair proteins, including PARP9, XRCC6, and TP53 [341], inducing DNA damage tolerance and TMZ resistance. Thus, HDAC6 inhibition may not only counteract GBM cells’ tumorigenic properties but also overcome resistance to therapies. Derailment of the autophagic pathway and inhibition of autophagosome–lysosome fusion have been observed as an action mechanism of HDAC6i. Interestingly, starvation-induced TDP-43, whose insoluble aggregates are associated with NDs such as amyotrophic lateral sclerosis, frontotemporal dementia (FTD), and ADs, activates autophagy in an HDAC6-dependent manner in GBM, blocking apoptosis. Consequently, HDAC6 inhibition reverses the TDP-43 autophagic process and the anti-apoptotic effect [342].

To date, three HDAC6i, namely ricolinostat, citarinostat, and KA2507, entered clinical trials for multiple myelomas [343] and solid cancers [316,317,318,344]. HDACi have been proven to be inefficient in synergising with the current therapeutic approaches for GBM and promote considerable side effects, and most of them do not cross the BBB. Thus, treatments able to block tumour progression or to induce sensitisation to radio- and chemotherapy are extremely urgent. Affecting different HDAC6 functions may be a useful strategy to target different GBM properties. The design of inhibitors that hit the diverse domains involved in the execution of the activities HDAC6 exert in GBM cells could greatly enhance selectivity, whereas potency could be powered by adducts targeting GBM subtype-specific molecular signatures while penetrating the BBB. Inclusion into nanomicelles or the design of piperazine-based HDAC6i [200,201] may pave the way to the production of more efficient brain-penetrative drugs. This strategy may foster the design of personalised therapies aimed first at increasing the life expectancy of GBM patients, who are currently without an effective cure.

## Figures and Tables

**Figure 1 biomedicines-12-02631-f001:**
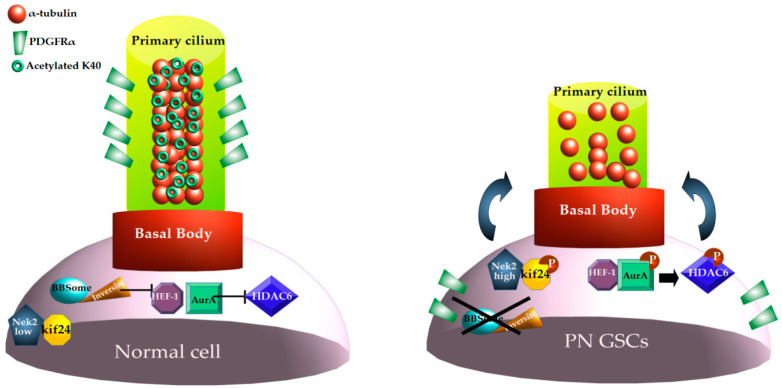
Schematic overview of primary cilia assembly/disassembly and players involved in normal and proneural (PN) glioblastoma stem cells (GSCs).

**Figure 2 biomedicines-12-02631-f002:**
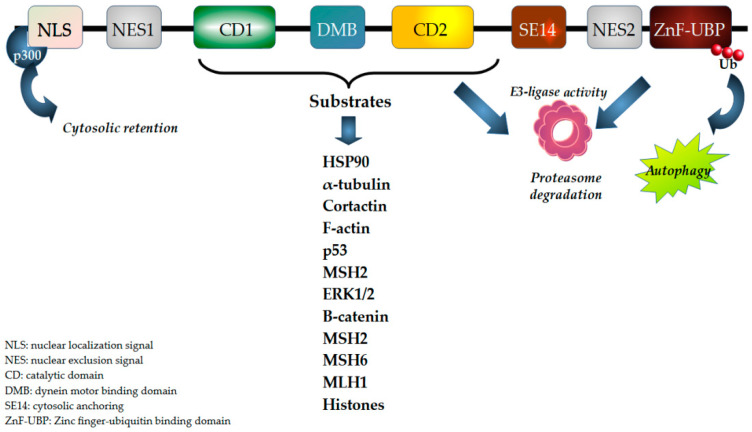
Multidomain HDAC6 structure is shown. Substrates and HDAC6 functions discussed in the present work are indicated.

**Figure 3 biomedicines-12-02631-f003:**
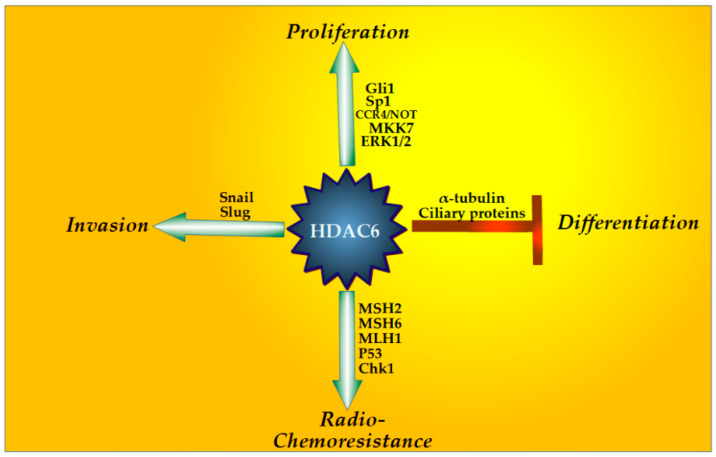
Pathways potentially affected by HDAC6 in GBM. Multifaceted HDAC6 may impact diverse processes pivotal to GBM ontogeny and progression, suggesting HDAC6 to be potentially targeted for therapeutic purposes.

## Data Availability

Not applicable.

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
