# Peer review of "The Role of HDAC6 in Glioblastoma Multiforme: A New Avenue to Therapeutic Interventions?"

_biomedicines, 2024, doi:10.3390/biomedicines12112631_

Round 1
Reviewer 1 Report
Comments and Suggestions for Authors
I reviewed and liked the article submitted by Spalloja and Illi, titled The Role of HDAC6 in Glioblastoma Multiforme: A New Avenue to Therapeutic Interventions?. The reason is that the article provides a comprehensive overview of the role of HDAC6 in glioblastoma multiforme (GBM), which is very relevant in therapeutic interventions.
I also realize that this study would immensely help in detailing the role of HDAC6 in key functions such as tumor cell proliferation. Besides, DNA repair, and chemoresistance. Moreover, it nicely combines intricate knowledge about epigenetics. More importantly, provides robust arguments to support the targeting of HDAC6 in therapeutic strategies.
However, I suggest some areas of improvement, including clarity concerning the definition of functional mechanisms in different subtypes of GBM based on HDAC6 functions. The paper would benefit significantly if information is included from recent clinical trials that support the therapeutic value of HDAC6 inhibitors. This would have practical relevance for adding to the usefulness of the article and strengthening the basis of the proposed therapeutic applications.
The overall I would say that with these edits, this article would serve as a good resource for the researcher who needs to review targeted therapies in GBM.
Author Response
biomedicines-3318905 Answers to Reviewers’ comments
First, the authors wish to thank both Reviewers for evaluating their work as valuable and informative.
Below, the answers to each Reviewer are reported.
Reviewer 1
Overall comment
I reviewed and liked the article submitted by Spalloja and Illi, titled The Role of HDAC6 in Glioblastoma
Multiforme: A New Avenue to Therapeutic Interventions?. The reason is that the article provides a
comprehensive overview of the role of HDAC6 in glioblastoma multiforme (GBM), which is very relevant in
therapeutic interventions.
I also realize that this study would immensely help in detailing the role of HDAC6 in key functions such as
tumor cell proliferation. Besides, DNA repair, and chemoresistance. Moreover, it nicely combines intricate
knowledge about epigenetics. More importantly, provides robust arguments to support the targeting of HDAC6
in therapeutic strategies……..The overall I would say that with these edits, this article would serve as a good
resource for the researcher who needs to review targeted therapies in GBM.
Author response
The authors deeply wish to thank the Reviewer for her/his positive evaluation of their work.
The authors believe to have fulfilled the Reviewer’s requests, as in the point to point answers below. Changes
to the MS according to this Reviewer’s comments are highlighted in light blue.
Q1. However, I suggest some areas of improvement, including clarity concerning the definition of functional
mechanisms in different subtypes of GBM based on HDAC6 functions.
A1. The authors thank the Reviewer for this helpful comment. An entire subparagraph has been added (new
subsection 4.3.5, lines 881-901). This subparagraph attempts to parallel GBM molecular subtypes and HDAC6
properties, providing a sort of “HDAC6-based” GBM classification.
Q2. The paper would benefit significantly if information is included from recent clinical trials that support the
therapeutic value of HDAC6 inhibitors.This would have practical relevance for adding to the usefulness of the
article and strengthening the basis of the proposed therapeutic applications.
A2. The authors really thank the Reviewer for raising this issue, covering a fundamental aspect not discussed
in the previous version of their MS. Also in this case, an entire subparagraph has been added (subpargraph
4.4.3, lines 978-1000), describing existing HDAC6i-based clinical trials and their outcome, when available.
Reviewer 2 Report
Comments and Suggestions for Authors
The article reviews the effects of HDAC6 expression on GBM and other diseases (such as renal fibrosis, autoimmune diseases, inflammatory diseases, etc.). It explains in detail the mechanisms by which HDAC6 promotes GBM cells from the perspectives of cell movement and migration, protein repair and degradation, and cell signal transduction. Furthermore, it suggests targeting HDAC6 for therapy and developing specific inhibitors to treat the aforementioned diseases. Although this research is valuable, it still has the following shortcomings:
1.The research on the current status of domestic and international research in Introduction is not sufficient, please add appropriately.
2.Part 3.4 only describes the existing research results and lacks a personal summary.
3.Part 4.1 mentions HDAC6 PROTACS with only a short statement and lacks a comprehensive review.
4. What are the advantages of HDAC6i over HDACi?
5.There are few references in the past five years. Please adjust the references appropriately to ensure the timeliness of the review.
Comments on the Quality of English LanguageThere are some spelling and grammar errors in the manuscript.
Author Response
biomedicines-3318905 Answers to Reviewers’ comments
First, the authors wish to thank both Reviewers for evaluating their work as valuable and informative.
Below, the answers to each Reviewer are reported.
Reviewer 2
Overall comment
The article reviews the effects of HDAC6 expression on GBM and other diseases (such as renal fibrosis,
autoimmune diseases, inflammatory diseases, etc.). It explains in detail the mechanisms by which HDAC6 promotes GBM cells from the perspectives of cell movement and migration, protein repair and degradation,and cell signal transduction. Furthermore, it suggests targeting HDAC6 for therapy and developing specific inhibitors to treat the aforementioned diseases.
Authors’ response
The authors wish to thank the Reviewer for the throrough analysis of their MS. Changes to the MS according to the Reviewer’s requests are highlighted in green.
A1. Although this research is valuable, it still has the following shortcomings. The research on the current status of domestic and international research in Introduction is not sufficient, please add appropriately.
Q1. The authors thank the Reviwer for this comment. Introduction section has been enriched accodingly (lines 86-102 and 107.116).
Q2..Part 3.4 only describes the existing research results and lacks a personal summary
A2. The authors thank the Reviewer for this comment. Indeed, the authors noticed that subparagraph 3.4 was lacking. Two subpargraphs 3.3 were present in v1. If the authors correctly interpreted the Reviewer’s request, the subsection discussing HDACi application in GBM has been implemented with personal statements related to the use of these epidrugs in human diseases, cancer and GBM (subparagraph 3.4, lines 501-511, lines 513- 514, lines 525-527).
Q3. Part 4.1 mentions HDAC6 PROTACS with only a short statement and lacks a comprehensive review.
A3. The authors apologize for the scarce discussion of subsection 4.4.1 and thank the Reviewer for raising this point. A deeper discussion of PROTACs has been provided in the revised version of the MS (lines 950-963).
Q4. What are the advantages of HDAC6i over HDACi?
A4. We thank the Reviewer for this comment. The answer to this question is provided in lines 998-1000 and 1129-1131.
Q5.There are few references in the past five years. Please adjust the references appropriately to ensure the timeliness of the review.
A5. The authors thank the Reviewer for this comment. We perfectly understand this Reviewer’s point.
However, we have to underlie that, when cited, dated papers either still represent a reference point for the scientific community working in the fields of research discussed in this review MS or discuss early observations regarding HDAC6 functions. When possible, references have been changed according to the Reviewer’s request.